# Supplementation with *Spirulina platensis* Prevents Uterine Diseases Related to Muscle Reactivity and Oxidative Stress in Rats Undergoing Strength Training

**DOI:** 10.3390/nu13113763

**Published:** 2021-10-24

**Authors:** Paula Benvindo Ferreira, Anderson Fellyp Avelino Diniz, Francisco Fernandes Lacerda Júnior, Maria da Conceição Correia Silva, Glêbia Alexa Cardoso, Alexandre Sérgio Silva, Bagnólia Araújo da Silva

**Affiliations:** 1Programa de Pós-graduação em Produtos Naturais e Sintéticos Bioativos, Universidade Federal da Paraíba, João Pessoa 58051-970, Brazil; paulabenvindo92@hotmail.com (P.B.F.); andersonfellyp@gmail.com (A.F.A.D.); lacerdafar17@gmail.com (F.F.L.J.); ceicafarma@gmail.com (M.d.C.C.S.); 2Departamento de Educação Física, Centro de Ciências da Saúde, Laboratório de Estudos de Treinamento Físico Aplicado à Performance e à Saúde, Universidade Federal da Paraíba, JoãoPessoa 58051-900, Brazil; gacbrasil@hotmail.com (G.A.C.); alexandresergiosilva@yahoo.com.br (A.S.S.); 3Departamento de Ciências Farmacêuticas, Centro de Ciências da Saúde, Universidade Federal da Paraíba, João Pessoa 58051-900, Brazil

**Keywords:** *Spirulina platensis*, physical exercise, uterus, oxidative stress, muscle reactivity

## Abstract

Strength training increases systemic oxygen consumption, causing the excessive generation of reactive oxygen species, which in turn, provokes oxidative stress reactions and cellular processes that induce uterine contraction. The aim of this study was to evaluate the possible protective effect of *Spirulina platensis* (SP), an antioxidant blue algae, on the contractile and relaxation reactivity of rat uterus and the balance of oxidative stress/antioxidant defenses. Female Wistar rats were divided into sedentary (CG), trained (TG), and T + supplemented (TG50, TG100) groups. Reactivity was analyzed by AQCAD, oxidative stress was evaluated by the malondialdehyde (MDA) formation, and the antioxidant capacity was measured by the 2,2-diphenyl-1-picrylhydrazyl (DPPH) method. Strength training increased contractile reactivity and decreased the pharmaco-mechanical component of relaxing reactivity in rat uterus. In addition, training decreased oxidation inhibition in the plasma and exercise increased oxidative stress in the uterine tissue; however, supplementation with algae prevented this effect and potentiated the increase in antioxidant capacity. Therefore, this study demonstrated that food supplementation prevents changes in reactivity and oxidative stress induced by strength training in a rat uterus, showing for the first time, that the uterus is a target for this exercise modality and antioxidant supplementation with *S. platensis* is an alternative means of preventing uterine dysfunction.

## 1. Introduction

Regular training has numerous beneficial effects on human health through the induction of homeostatic adaptations in different physiological systems such as the cardiorespiratory and muscle systems [1]. However, the magnitude of the effect of a specific training regime can vary significantly between individuals, as well as in individuals undergoing training who may not respond as expected [2]. This is due to factors such as the characteristics of the training regime, environmental conditions and individual factors, such as habitual physical activity, previous physical fitness level, genetics, psychological factors, age and sex [3].

In the past few decades, women have become increasingly physically active and evidence demonstrates that physical training can increase self-esteem, cardiorespiratory fitness, ovulation, and menstrual regularity while decreasing insulin resistance and body fat [4]. Despite this, several studies have reported that the female reproductive system is highly sensitive to physiological stress, and training with excessive loads is related to diseases such as osteoporosis and reproductive disorders including late menarche, primary and secondary amenorrhea and oligomenorrhea, which occur in 6% to 79% of women involved in athletic activities [5,6].

Additionally, free radicals have a dual role in the reproductive system; they function as key signaling molecules that modulate various reproductive functions and can directly influence the quality of oocytes, the oocyte–sperm interaction, the implant and initial embryonic development in their microenvironments [7]. The extent to which reactive species are useful or harmful is determined by factors such as the duration of exercise, the intensity, physical conditioning and the nutritional status of the individual [8]. Thus, the uterus should definitely be considered an important organ, making it an attractive target for exercise.

Oxidative stress has been widely studied within the scope of aerobic exercise. However, responses to anaerobic exercise, such as strength training have hardly been explored and the main focus has been on the muscles involved with exercise while the consequences for other types of muscles have been neglected. Thus, it is important to investigate the changes in the smooth muscles as a result of the practice of physical exercise since these muscles are mainly responsible for the control of most of the hollow organs in the body systems, including the uterus [9]. Different stages of molecular mechanisms related to muscle contraction have been shown to be susceptible to redox modulation, including the regulation of Ca^2+^ channels in the sarcoplasmic reticulum and myofibrillar sensitivity to Ca^2+^ [10,11,12].

Considering the important role of oxidative stress in the pathophysiology of several diseases, including some uterine disorders, and that poorly managed training loads can promote oxidative stress, there has recently been an increase in the consumption of nutrients that can act beneficially, either in isolation or in association with physical training, to improve the redox balance [13]. Food intake or antioxidant supplements are used as a non-invasive tool to decrease muscle damage, improve exercise performance, prevent or reduce oxidative stress, and improve lifespan with fewer of the specific risks that strenuous exercise can cause in athletes [14,15]. The benefits of antioxidant supplements may be related to an improvement in the cellular redox state, and in turn, a decrease in oxidative changes in DNA, lipids and proteins [16].

In this context, marine organisms, especially algae-derived compounds, play an important role among natural products due to the presence of secondary metabolites with great chemical diversity [17]. One marine organism that deserves to be highlighted is *Spirulina platensis* (Oscillatoriaceae), a blue-green algae that has been attracting attention due to its medicinal and nutritional properties, because of its antioxidant properties [18] and its effects on the smooth muscles of the aorta [19], trachea [20], ileum [21] and cavernous body [22].

Therefore, experimental models of strength training are considered to be a viable way to investigate the effects of organic dysfunctions induced by exercise on uterine contractility.Assuming that algae promotes beneficial effects in models of intestinal smooth muscle [23] and the vascular system [24] of animals submitted to intense strength training, it was hypothesized that strength exercise alters the contractile and relaxing reactivity of the Wistar rat uterus, and that food supplementation with *S. platensis* prevents uterine dysfunctions caused by exercise.

## 2. Materials and Methods

### 2.1. Substances

The salts used for the Locke Ringer’s solution were purchased from Vetec (Rio de Janeiro, Brazil), Nuclear (Porto Alegre, Brazil) and Dinâmica (Diadema, Brazil). MDA, 1,1-diphenyl-2-picryl hydrazil (DPPH), methane hydroxymethylamine (tris), phenyl-methyl-sulfonyl fluoride (PMSF), aprotinin, dithiothreitol (DTT), Tween 20 and albumin were purchased from Sigma Aldrich (Rio De Janeiro, Brazil). Distilled water was used to dilute the substances and prepare the stock solutions and the diethylstilbestrol was dissolved in absolute alcohol (96° GL). The carbogen mixture (95% O_2_ and 5% CO_2_) was obtained from White Martins (Rio De Janeiro, Brazil). All substances were weighed on analytical scales, GEHAKA model AG 200 (São Paulo, Brazil).

### 2.2. Animals

Two-month-old virgin Wistar female rats (*Rattusnorvegicus*) weighing approximately 150–250 g, were purchased from the Animal Production Unit of the Instituto de Pesquisaem Fármacos e Medicamentos (IPeFarM). The animals were maintained under controlled ventilation and temperature (21 ± 1 °C) with water ad libitum in a 12 h light-dark cycle (lights on from 6 a.m. to 6 p.m.). Female rats were treated 24 h before euthanasia with diethylstilbestrol (1 mg/kg, s.c.) for estrus induction. The experiments were carried out from 7 a.m. to 11 p.m. The euthanasia of the rats was performed during the light period of this cycle. The experimental procedures (Ethics Committee on Animal Use of UFPB: 0211/2014) were performed following the guidelines for the ethical use of animals in applied etiology studies [25], and those of the National Council for Animal Experimentation Control (in Brazil) [26].

### 2.3. Preparation and Supplementationwith Spirulina platensis

*Spirulina platensis* in powder form was purchased from the INFINITY Pharma laboratory (Hong Kong, China) (Lot No. 17J11-B004-02504). The powder was prepared at the Roval Manipulation Pharmacy (João Pessoa, Brazil) (Lot No. 20121025). The *S. platensis* powder was prepared and dissolved daily in saline solution (NaCl 0.9%) and a solution was obtained at the dose to be used in the study (50 and 100 mg/kg) and administered to the rats after its preparation [19]. Supplementation occurred for a period of eight weeks [27] and algae was administered orally, thirty minutes before the exercise session [28] with the aid of stainless-steel needles for gavage (BD-12, Insight, Ribeirão Preto, Brazil) and 5 mL disposable syringes with 0.2 mL precision (BD, João Pessoa, Brazil).

### 2.4. Experimental Groups

Female rats were randomly divided into a sedentary group, or submitted to a strength training protocol and supplemented with *S. platensis* (50 and 100 mg/kg). Thus, the study consisted of the following groups with 28 rats each: sedentary group (CG, control), a group trained for 8 weeks (TG), a group trained and supplemented with algae 50 mg/kg (TG50), and a group trained and supplemented with *S. platensis* 100 mg/kg (TG100).

### 2.5. Strength Training Program

Female rats belonging to the strength training group were submitted to a training program that consisted of jumping in a liquid medium in a cylindrical PVC container (dimensions: 30 cm in diameter and 70 cm in length). The depth of water in the tanks was 50 cm, equivalent to twice the length of the mouse to prevent them from climbing to the edge of the cylinder. The water was previously heated to a temperature of 32 °C, as this was considered neutral in relation to the rat’s body temperature [29].

Strength training was performed according to the protocol developed by Marqueti et al. (2006) for jumping in liquid medium [29]. The protocol consisted of 4 sets of 10 to 12 repetitions with an interval of 30 s between sets, and with progressive overload being adjusted according to the animal’s weight (Figure 1). The overload was applied to the animals’ chest through a fabric vest that allowed the jumps to be performed without the load disconnecting from the body or impeding their movements. An overload corresponding to the weight of the vest when wet (25 g) was considered and charged to the specific load corresponding to the animal’s body mass for better training accuracy.

After the end of the training protocols, the animals in the training group were euthanized by anesthesia with ketamine 100 mg/kg (i.p.) and xylazine 10 mg/kg (i.p.), followed by a complementary method of decapitation with the aid of a guillotine. They were euthanized 48 h after the last exercise session in order to eliminate the acute effect of the exercise on the reactivity (MOURA et al., 2012) [30].

### 2.6. Isolating the Uterus of Female Rats

Female rats were treated 24 h before euthanasia with diethylstilbestrol (1 mg/kg, s.c.) for estrus induction. The abdominal cavity was opened, and the uterus was dissected and placed in Locke Ringer’s nutrient solution at 32 °C gassed with a mixture of carbogen. Then, the 2 uterine horns were separated by incision, opened longitudinally and suspended vertically by cotton thread in baths of isolated organs (6 mL), under tension of 1 g and kept at rest for at least 40 min. The solution was changed to every 15 min [31].

Uterus and plasma fragments were obtained for the biochemical measurements. These samples were quickly removed and stored in a freezer at −80 °C until the moment of analysis.

Locke Ringer’s solution (adjusted to pH 7.4 with NaOH or 1N HCl) was carbonated with carbon and kept at 32 °C, and its composition (mM) was: NaCl (154.0); KCl (5.6); CaCl_2_ (2.2); MgCl_2_ (2.1); glucose (5.6); NaHCO_3_ (6.0).

### 2.7. Contractile Reactivity Evaluation

As described above, the uterus was assembled and after the stabilization period (40 min), a contraction with 60 mMKCl was induced to verify the organ’s functionality. After 15 min, two cumulative concentration–response curves for KCl or oxytocin were obtained. The contractile reactivity was calculated from the maximum contraction of the uterus of the animals from the groups that received supplementation with *S. platensis* and/or were submitted to strength training, being that obtained by the average of the maximum amplitudes of the control curves. Comparisons were made between groups that received supplementation with *S. platensis* and/or underwent strength training, using maximum effect values (Emax) and the negative logarithm (base 10) of the concentration of KCl or oxytocin producing 50% of the Emax (pEC_50_).

### 2.8. Relaxation Reactivity Evaluation

After the stabilization period, a uterine contraction was induced with oxytocin (10^−2^ IU/mL) or KCl (60 mM). After the formation of the tonic component, isoprenaline or nifedipine was added cumulatively to the organ bath of all groups, in different preparations [32]. The relaxation response was observed and expressed as the reverse percentage of the initial contraction produced by the contractile agents. Comparisons were made between groups that received supplementation with algae and/or underwent strength training, with the means of the maximum amplitudes of the control curves, based on the Emax and pEC_50_ values of the relaxation substances, being calculated from the cumulative concentration–response curves that were obtained.

### 2.9. Evaluation of Oxidative Stress (MDA) and Antioxidant Defenses (CAT)

After the animals were euthanized, the cervical vessels were sectioned to collect blood in test tubes with anticoagulant (EDTA) and were centrifuged at 1198× *g* for 10 min. The supernatant was stored in Eppendorf^®^ tubes and refrigerated at 20 °C [33,34]. To obtain the homogenates, the uterine horns were isolated and frozen at 20 °C. The tissue was weighed, macerated and homogenized with 10% KCl at a ratio of 1:1. Afterwards, the samples were centrifuged (1198× *g*/10 min) and the supernatant was separated for testing.

Lipid peroxidation was evaluated by quantifying the MDA production. This was performed using the method of quantification of thiobarbituric acid reactive species (TBARS) in which two TBA molecules condense with one MDA molecule through a reaction colorimetric; this product can be easily detected by spectrophotometry. After obtaining plasma and uterine homogenate, 250 µL aliquots were incubated at 37 °C in a water bath for 60 min. Afterwards, samples were precipitated with 400 µL of 35% perchloric acid and centrifuged at 16,851× *g* for 20 min at 4 °C. The supernatant was transferred to Eppendorf^®^ tubes, 400 µL of thiobarbituric acid (TBA) 0.6% was added to the samples and incubated at 95–100 °C for 1 h. Then, after cooling, the samples were read in a spectrophotometer at 532 nm. The turbid samples taken after the water bath were centrifuged again at 1198× *g* for 10 min before reading [35].

The colorimetric method of the reduction of DPPH was used to assess the total antioxidant capacity (CAT).This method is based on the ability of the sample to reduce the DPPH radical, which has a purple color, to 1,1-diphenyl-2-picryl hydrazine, a colored transparent, which is detected by spectrophotometry. Thus, 50 µL of plasma or pulmonary homogenate and 2 mL of DPPH solution dissolved in absolute ethanol (0.012 g/L) were added to a centrifuge tube and protected from light; then, the tubes were vortexed for 10 s and kept at rest for 30 min. Next, the samples were centrifuged at 7489× *g* for 15 min at 20 °C. The supernatant was read in a spectrophotometer at 515 nm. [36]

Analyses was performed to compare the MDA levels (μmol/L of sample) or the CAT (%) between the CG, TG, TG50 and TG100 groups.

### 2.10. StatisticalAnalysis

The functional results obtained were expressed as mean and standard error of the mean (S.E.M.) (*n* = 5) and statistically analyzed for intergroup comparison using Student’s *t*-test. The results were statistically analyzed using two-way analysis of variance (ANOVA) followed by Tukey’s post-test. The differences between the means were considered significant when *p* < 0.05. The pCE_50_ values were calculated using linear regression, and *E*_max_ was obtained by averaging the maximum percentages of contraction or relaxation. All results were analyzed using Graph Pad Prism version 5.01 (Graph Pad Software Inc., San Diego, CA, USA).

## 3. Results

### 3.1. Effects of Strength Training and Supplementation with S. platensis on the Contractile Reactivity of the Uterus to KCl and Oxytocin

In the group submitted to strength training (GT), there was a reduction in potency and an increase in contractile efficacy in response to KCl (pCE_50_ = 1.0 ± 0.03; E_max_ = 172.7 ± 8.1%) when compared to the CG (pCE_50_ = 2.0 ± 0.07; E_max_ = 100%). Supplementation with *S. platensis* in GT50 (pCE_50_ = 1.6 ± 0.02; E_max_ = 84.3 ± 8.8%) and in GT100 (pCE_50_ = 2.1 ± 0.05; E_max_ = 119.7 ± 9.1%) prevented the increase in efficiency and reduced the potency of KCl (Figure 1a).

In female rats submitted to strength training, there was a reduction in potency and an increase in the contractile efficacy of oxytocin (pCE_50_ = 2.1 ± 0.1; E_max_= 222.0 ± 7.1%) compared to the CG (pCE_50_ = 3.4 ± 0.2; E_max_ = 100.0%) and food supplementation with *S. platensis* in GT50 (pCE50 = 3.5 ± 0.1; E_max_ = 207.2 ± 15.0%) and in GT100 (pCE_50_ = 3.5 ± 0.1; E_max_ = 169.1 ± 7.7%) partially prevented the increase in efficiency and reduced the potency of this agonist (Figure 1b).

### 3.2. Effect of Strength Training and Supplementation with S. platensis on the Relaxing Response in Utero to Nifedipine and Isoprenaline

Strength training did not alter the potency or relaxing efficacy of nifedipine (pCE_50_ = 11.0 ± 0.2; E_max_ = 100%) compared to the CG (pCE_50_ = 10.6 ± 0.08; E_max_ = 100%). However, dietary supplementation with seaweed in the GT50 (pCE_50_ = 8.8 ± 0.2; E_max_ = 100%) and in the GT100 (pCE_50_ = 8.6 ± 0.2; E_max_ = 100.0%), decreased the relaxing potency of nifedipine, with no change in efficacy (Figure 2a).

Strength training (E_max_ = 89.6 ± 3.6%) decreased the relaxing efficacy of isoprenaline in relation to CG (E_max_ = 100%) and supplementation with algae in GT50 and GT100 (E_max_ = 100%) prevented the decrease in effectiveness. In relation to the potency, the GT (pCE50 = 9.8 ± 0.3) decreased the relaxing power of isoprenaline compared to the CG (pCE_50_ = 12.0 ± 0.3); food supplementation with *S. platensis* prevented this reduction in both doses (GT50–pCE_50_ = 12.0 ± 0.2; GT100–pCE_50_= 12.3 ± 0.2) (Figure 2b).

### 3.3. Effect of Strength Training and/or Food Supplementation with S. platensis on the Concentration of MDA in the Plasma and Uterus of Rats

In female rats submitted to strength training and supplemented with *S. platensis* at doses of 50 and 100 mg/kg, no difference was observed in the plasma MDA concentration of the TG (3.6 ± 0.2), TG50 (3, 8 ± 0.4) and TG100 (3.1 ± 0.1) compared to the control CG (3.2 ± 0.1) (Figure 3a).

When analyzing the samples of uterine tissue, the concentration of MDA was increased in the TG group (3.9 ± 0.1) and this increase was prevented by supplementation with algae in the TG50 groups (1.9 ± 0.2) and more markedly in TG100 (1.0 ± 0.05) compared to the control CG (1.6 ± 0.1) (Figure 3b).

### 3.4. Effect of Strength Training and/or Food Supplementation with S. platensis on the Total Antioxidant Capacity in Plasma and Rat Uterus

In rats trained and supplemented with *S. platensis*, it was shown that strength training decreased the percentage of inhibition of oxidation in plasma (15.2 ± 0.4%) and that supplementation with seaweed in TG50 (22.4 ± 0.1%) and TG100 (24.6 ± 0.9%) attenuated this effect compared to the CG (28.4 ± 0.7%) (Figure 4a).

In the uterus, the oxidation percentage was increased by TG strength training (97.4 ± 1.3%) and by supplementation with *S. platensis* in TG50 (107.4 ± 1.9%), with this increase being marked in TG100 (130.8 ± 4.0%) compared to the CG (83.6 ± 3.4%) (Figure 4b).

## 4. Discussion

In this research, we investigated the modulation of contractile reactivity in the uterus of rats by strength training and in the association of dietary supplementation with training, as well as its effects on the balance of oxidative stress and antioxidant defenses. As a result, we demonstrated that strength training decreases potency and increases contractile efficacy, and decreases relaxing potency in rats’ wombs, and these effects were prevented by dietary supplementation with *Spirulina platensis*. In addition, we found that supplementation with kelp prevented the increase in oxidative stress and improved uterine antioxidant defenses in trained female rats.

Several studies have shown that the pathophysiology of diseases that affect organs covered by smooth muscle, such as the intestine, vessels and uterus, may involve the deregulation of oxidative stress, as this is harmful to the contraction of smooth muscle. In addition it promotes changes in the function of various proteins from receptors to ion channels, which are responsible for triggering contractile processes [37]. The reactivity of the uterine smooth muscle plays a critical role in regulating and controlling the contractile activity of the myometrium. Many problems such as abortions, premature births, postpartum hemorrhages and uterine colic are associated with the abnormal regulation of the contractility of this muscle [38,39,40].

It is also important to emphasize that when women who practice physical exercise are subjected to excessive exercise, which is often accompanied by inadequate recovery, this can lead to disturbances in the body’s homeostasis and hormonal dysregulation, which, in turn, can cause disorders in the reproductive system related to oxidative stress [5].

The correlation between oxidative stress and diseases involving the female reproductive system has led to an increase in the consumption of antioxidant supplements as a useful non-invasive tool, especially by women who exercise, and are seeking to decrease muscle damage, improve exercise performance and prevent or reduce oxidative stress. In addition, the concomitant use of these supplements has been shown to be promising and important for obtaining the positive results of the exercise [41]. In this scenario, *Spirulina platensis* stands out, and it is used by athletes and physical activity practitioners due to its antioxidant potential and its high protein concentration [42].

From this, it was hypothesized that strength training promotes deleterious changes in uterine contractile and relaxing reactivity by increasing oxidative stress and that dietary supplementation of rats with *Spirulina platensis* would prevent the effects of exercise on the uterine reactivity of rats by decreasing oxidative stress.

Uterine contractions happen through pharmaco-mechanical and electromechanical coupling, which predominantly occurs through an increase in cytosolic calcium concentration ([Ca^2+^]_c_) [43], with pharmaco-mechanical coupling being induced by the release of calcium (Ca^2+^) from intracellular cells mediated by inositol 1,4,5-triphosphate (IP3),while electromechanical contraction coupling is related to the change in resting potential, which promotes membrane depolarization, triggering the influx of Ca^2+^ through voltage-dependent calcium channels (Ca_V_), and consequently, a contractile response. Thus, oxytocin, an agonist of oxytocin receptors (OT), was used as a pharmaco-mechanical contractile agent and to simulate the myogenic of this musculature [44]. KCl was used as a depolarizing agent of the cell membrane, resulting in an influx of Ca^2+^ and contraction.

Our data showed that strength training induced an increase in contractile efficacy in the two investigated couplings, demonstrating that uterine contractile reactivity is a target for exercise, and that dietary supplementation with *Spirulina platensis* prevented these effects in all tested doses. These results indicate that *Spirulina platensis* acts to prevent the formation of contractile compounds resulting from the practice of strength training, which in turn reduces the influx of Ca^2+^ and promotes a decrease in the contractile reactivity of the uterine smooth muscle.

Brito (2014) [28] demonstrated that supplementation with *S. platensis* potentiates the relaxing reactivity of the aorta (at doses of 150 and 500 mg/kg), against acetylcholine, and a decrease in contractile reactivity towards phenylephrine in the aorta of healthy rats. Similarly, Souza (2018) [45] found that supplementation with *S. platensis* potentiated the relaxing reactivity of the cavernous body of a rat. Thus, it was hypothesized that food supplementation with *S. platensis* would alter the relaxing reactivity of a rat uterus.

With regard to the relaxing response using nifedipine, an electromechanical agent Ca_V_ blocker, it was observed that strength training is not related to that component of the relaxing response. In the evaluation of the pharmaco-mechanical relaxation coupling, it was observed that the effect promoted by isoprenaline, a pharmaco-mechanical agonist, once again demonstrated the preventive effect of *S. platensis,* since the reduction in the relaxing potency was prevented in all doses tested. These data corroborate those observed in contractile reactivity and indicate that *Spirulina platensis* might target specific points that have been modulated by exercise.

In view of the data presented, it appears that the strength training model promoted changes in the contractile and relaxing uterine reactivity, with a resulting increase in contractile function, and that these changes were prevented by supplementation with *S. platensis*. It is known that strength exercise is strongly associated with the generation of ROS that alter tissue homeostasis, leading to different adaptations of the physiological changes induced by the deviation in cardiac output and hypoxia, which can affect the female reproductive system [5]. These adjustments are reflected by changes in contractile proteins, mitochondrial function, metabolic changes, intracellular signaling and transcriptional responses [46]. Thus, it was postulated that strength training and supplementation with *S. platensis* would modify the oxidative stress/antioxidant defense balance in plasma.

For this study, we used a methodology that determines MDA levels through a colorimetric reaction that results in the formation of a fluorescent pink chromogen that can be detected by means of a spectrophotometer reading [47], to confirm or discard the hypothesis that food supplementation with *S. platensis* reduces the formation of free radicals and prevents oxidative stress induced by strength training in the uterus of female rats Wistar.

When analyzing the rats submitted to the training protocol, an increase in MDA levels was observed, which was prevented by supplementation with seaweed in the tested doses. Although a sharp increase in the production of oxidants is necessary for myocellular adaptations to exercise, a chronic increase in protein oxidation can have serious consequences.

The chronic response to both aerobic and anaerobic training is to improve the antioxidant capacity of plasma, and the acute response relates to the production of specific antioxidants [48]. Based on this premise and that oxidative damage stimulates antioxidant defenses, we evaluated whether food supplementation with *S. platensis* and strength training would alter the systemic and tissue antioxidant capacity of Wistar rats [49].

When we analyzed the groups submitted to strength training, a decrease in the percentage of plasma oxidation inhibition was observed, demonstrating that strength training can decrease the systemic antioxidant capacity and that supplementation with *S. platensis* partially prevented this decrease. Despite this, there were no consequences related to systemic lipid peroxidation. However, when analyzing this parameter at the tissue level, there was an increase in antioxidant capacity in all groups tested, indicating that *S. platensis* potentiates the increase in antioxidant capacity in rats’ uterus induced by strength training, strongly suggesting that a decrease in contractile free radicals resulting from the practice of strength exercise is one of the possible mechanisms to explain this preventive effect.

It is already well-established that pro-oxidant and antioxidant factors interact in a complex way to reach levels that do not damage the intracellular environment. When this balance is disturbed by an increase in free oxidizing substances, oxidative stress occurs and this event affects the whole body [19]. Such conditions are related to muscle fatigue [50], diabetes [51], arterial hypertension [52] and intermittent ischemia induced by myometrium contractions that restrict blood flow to the uterus, causing a state of reperfusion/ischemia [53,54].

Therefore, when physical exercise is not prescribed correctly, ROS can induce lipid peroxidation leading to problems such as inactivation of cell membrane enzymes, necrosis of muscle fibers, release of cellular enzymes into the blood, decreased effectiveness of immune system and alteration of mitochondrial function, leading to decreased muscle performance, overtraining and impairing important adaptations to training [55].

Thus, one can suggest that dietary supplementation with *S. platensis* has promising potential in pathophysiological processes that involve dysregulations in uterine contractile homeostasis such as dysmenorrhea, premature birth and abortion, as well as its preventive action for women who practice intense physical activity.

## 5. Conclusions

Based on the data presented and the discussion above, it was demonstrated, for the first time, a beneficial preventive effect of dietary supplementation with *S. platensis*, as well as its association with strength training.These preliminary data provide new insights into the mechanisms involved in this effect, and require tests in humans to confirm the possible effects, such as the reduction in uterine oxidative damage, which may contribute to better uterine functioning, especially for practitioners of progressively intense exercise during exposure to stress.

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
