# Peer review of "Supplementation with *Spirulina platensis* Prevents Uterine Diseases Related to Muscle Reactivity and Oxidative Stress in Rats Undergoing Strength Training"

_nutrients, 2021, doi:10.3390/nu13113763_

Round 1

Reviewer 1 Report

Comments and suggestions for authors:

The author indicates that Spirulina platensis has antioxidant potential, which not only delays the oxidative stress generated by the body during intense exercise, but also provides the high protein concentration required by sports practitioners. They found that dietary supplementation of rats with Spirulina platensis would prevent the effects of exercise on the uterine reactivity of rats by decreasing oxidative stress. The purpose of this article is clear and interesting, but there are still some questions as follows:

  1. The author mentioned that the strength training model causes uterine contraction homeostasis in rats, but for female athletes suffering from dysmenorrhea or premature birth, will high-intensity exercise be the main cause of uterine contraction disorders? How does the intensity of strength training in animal mode compare to the intensity of human exercise?
  2. In figure 3, the subtitle of the x-axis is wrong, please correct it.
  3. In figure 4, why does strength training reduce the systemic antioxidant capacity, but not the tissue level?
  4. Since the potential of antioxidant of Spirulina platensis is the main purpose of this research, if possible, I think more data should be supplemented, such as the ROS level, mitochondrial function, antioxidant-related protein expression, and downstream intracellular signaling.

Author Response

Response to Reviewer 1 Comments

Point 1: The author mentioned that the strength training model causes uterine contraction homeostasis in rats, but for female athletes suffering from dysmenorrhea or premature birth, will high-intensity exercise be the main cause of uterine contraction disorders? How does the intensity of strength training in animal mode compare to the intensity of human exercise? 

Response 1: High-intensity exercise induces biochemical and molecular alterations that can lead to hormonal disturbances in the hypothalamic-pituitary-ovarian axis and directly in the contractile machinery, which in turn can result in pathological damage, such as dimenorrhea and premature abortion. That is why it is important to know the changes that certain training models cause in the female reproductive system, in order to adapt the intensities of training protocols, as well as the use of supplementary substances to avoid them. The intensity of strength training used in this experimental model in rats is compared to the intensity of strength training in humans due to the experimental design of progressive strength training, which is based on an increase every two weeks in the number of jumps and/or load , which gives the training the maintenance of intensity without a natural adaptation.

Point 2: In figure 3, the subtitle of the x-axis is wrong, please correct it.

Response 2: X-axis correction performed.

Point 3: In figure 4, why does strength training reduce the systemic antioxidant capacity, but not the tissue level?

Response 3: Physical training is an event that challenges the homeostasis of the body as a whole as well as the antioxidant system. However, the antioxidant system of different body tissues present different responses to this challenge. In the evaluation carried out in plasma, there is an overview of the sum of the responses of all tissues in the body and in the tissue evaluation, the individual effect of that tissue is observed, which may be different because each tissue presents different sensitivity to the presented stimulus.

Point 4: Since the potential of antioxidant of Spirulina platensis is the main purpose of this research, if possible, I think more data should be supplemented, such as the ROS level, mitochondrial function, antioxidant-related protein expression, and downstream intracellular signaling.

Response 4: In this work, we revealed for the first time that progressive strength training alters uterine homestasis both in contractile and relaxing responses in pharmaco and electromechanical couplings, as well as in the redox state and that supplementation with Spirulina platensis prevents these alterations. Based on these discoveries, new experiments are being designed and carried out for an in-depth investigation of the mechanisms related to these findings, such as evaluation of the function of cyclooxygenase, NO synthase, NADPHoxidase, superoxide dismutase, catalase and MAPKs for future publications.

Reviewer 2 Report

In their study, the Authors investigated a possible protective effect of SP on contractile and relaxation reactivity of rat uterus and the balance of oxidative stress/antioxidant defenses. The study population including female Wistar rats were divided into sedentary (CG); trained (TG); and T + supplemented (TG50, TG100). Of interest, they found the following findings: a) strength training increased contractile reactivity and decreased the pharmaco mechanical component of rat uterus relaxing reactivity; b) training decreased oxidation inhibition in the plasm; c) in the uterine tissue, the exercise increased oxidative stress and supplementation with algae prevented this effect and potentiated the increase in antioxidant capacity. Therefore, the Authors concluded for the first time that uterus is a target for this exercise modality and the antioxidants supplementation with S. platensis could be an alternative to prevent uterine dysfunctions.

The study has the merit to stimulate the research on the topic of antioxidants supplementation as potential treatment of disease in obstetrics and gynecology. The methodology is well-reported and for this reproducible. Statistical analysis is appropriate, reference updated and English good.

In my opinion, the little sample size is a limitation, and it should be discussed in the discussion section. At same time, in obstetrics, other conditions could be managed by SP, such as gestational diabetes and hypertensive disorders, after demonstrating a safety profile of SP

Author Response

Response to Reviewer 2 Comments

Point 1: In their study, the Authors investigated a possible protective effect of SP on contractile and relaxation reactivity of rat uterus and the balance of oxidative stress/antioxidant defenses. The study population including female Wistar rats were divided into sedentary (CG); trained (TG); and T + supplemented (TG50, TG100). Of interest, they found the following findings: a) strength training increased contractile reactivity and decreased the pharmaco mechanical component of rat uterus relaxing reactivity; b) training decreased oxidation inhibition in the plasm; c) in the uterine tissue, the exercise increased oxidative stress and supplementation with algae prevented this effect and potentiated the increase in antioxidant capacity. Therefore, the Authors concluded for the first time that uterus is a target for this exercise modality and the antioxidants supplementation with S. platensis could be an alternative to prevent uterine dysfunctions.

The study has the merit to stimulate the research on the topic of antioxidants supplementation as potential treatment of disease in obstetrics and gynecology. The methodology is well-reported and for this reproducible. Statistical analysis is appropriate, reference updated and English good.

In my opinion, the little sample size is a limitation, and it should be discussed in the discussion section. At same time, in obstetrics, other conditions could be managed by SP, such as gestational diabetes and hypertensive disorders, after demonstrating a safety profile of SP

Response 1: The sample size was defined based on national regulatory standards for animal testing of the National Council for Animal Experimentation Control (in Brazil).

This manuscript is a resubmission of an earlier submission. The following is a list of the peer review reports and author responses from that submission.